# Self-Supervised Bisimulation Action Chunk Representation for Efficient RL

**Lei Shi, Jianye Hao, Hongyao Tang, Zibin Dong, Yan Zheng**
College of Intelligence and Computing, Tianjin University
{leishi, jianye.hao, bluecontra, yanzheng}@tju.edu.cn
zibindong@outlook.com

## Abstract

Action chunking in reinforcement learning is a promising approach, as it significantly reduces decision-making frequency and leads to more consistent behavior. However, due to inherent differences between the action chunk space and the original action space, uncovering its underlying structure is crucial. Previous works cope with this challenge of single-step action space through action representation methods, but directly applying these methods to action chunk space fails to capture the semantic information of multi-step behaviors. In this paper, we introduce **A**ction **C**hunk **R**epresentation (**ACR**), a self-supervised representation learning framework for uncovering the underlying structure of the action chunk space to achieve efficient RL. To build the framework, we propose the action chunk bisimulation metric to measure the principled distance between action chunks. With this metric, ACR encodes action chunks with a Transformer that extracts the temporal structure and learns a latent representation space where action churns with similar bisimulation behavior semantics are close to each other. The latent policy is then trained in the representation space, and the selected latent action chunk is decoded back into the original space to interact with the environment. We flexibly integrate ACR with various DRL algorithms and evaluate it on a range of continuous manipulation and navigation tasks. Experiments show that ACR surpasses existing action representation baselines in terms of both learning efficiency and performance.

## 1 Introduction

Action chunking reinforcement learning (RL) agents learn a policy that produces a sequence of actions at each step to solve tasks, which has been proven to reduce interaction frequency and lead to more consistent behavior [1, 2, 3, 4, 5, 6]. A common approach is to use the Cartesian product of the original action space as the action chunk space [6]. While straightforward, this approach leads to an exponential increase in dimensionality and introduces meaningless action combinations [7], making policy optimization challenging. Since only a few action combinations yield meaningful behaviors, learning the underlying structure of the action chunk space is crucial. While action representation methods [8] have shown promise in handling complex action spaces, they often focus on single-step action space settings, constructing representation spaces through forward prediction [9, 10, 11] or reconstruction [12, 7].

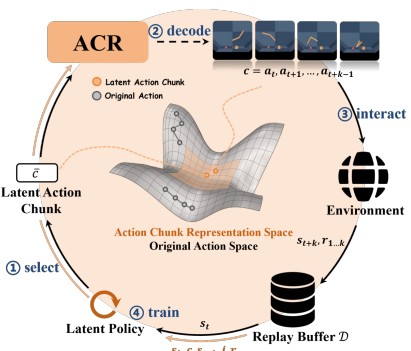

Figure 1: **Overview.** The latent policy is trained in the ACR action chunk representation space guided by the bisimulation metric. The selected latent action chunk is decoded back into the original space to interact with the environment.

38th Conference on Neural Information Processing Systems (NeurIPS 2024).

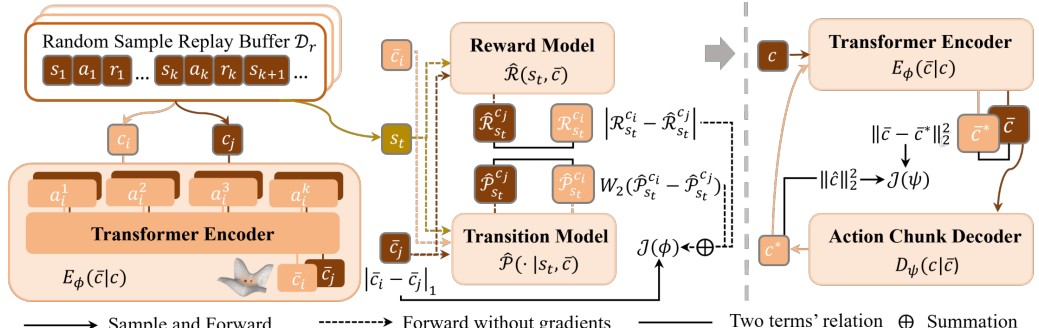

**Figure 2: The training process of ACR. Left:** We train the Transformer encoder to construct the action chunk representation space, ensuring that the $\ell_1$ distance between any two latent action chunks equals the distance measured by the ACB Metric. During training, we sample pairs $(s_t, c_i, \mathcal{R}_{s_t}^{c_i})$ and additional chunks $c_j$ from random samples. The encoder first generates latent action chunks $\bar{c}_i$ and $\bar{c}_j$, after which the transition model and reward model predict the transitions for $(s_t, c_i)$ and $(s_t, c_j)$, and the cumulative reward for $(s_t, c_j)$. **Right:** We train the action chunk decoder once the encoder is fully trained. The decoded action chunk $c^*$ is then passed through the fixed encoder to obtain $\bar{c}^*$.

Unfortunately, directly using single-step representation methods in the action chunk space still introduces the curse of dimensionality and fails to capture the semantic information of multi-step behavior, largely due to the inherent underlying structure. Therefore, we aim to develop a behavior-centric self-supervised representation method that is inherently suitable for action chunk space.

In this paper, we propose a novel self-supervised representation learning framework for action chunk space, called **A**ction **C**hunk **R**epresentation (**ACR**), which constructs a compact and decodable action chunk representation space, enabling efficient policy learning and interaction with the environment. An overview of ACR is shown in Fig.1. In contrast to commonly used forward prediction or reconstruction objectives, we introduce a novel action chunk bisimulation metric as the distance metric for constructing the ACR representation space, thereby regularizing the learned representation to focus on the behavior influence of action chunks. Moreover, ACR utilizes a Causal Transformer [13] encoder to capture the temporal information within action chunks, which is crucial for constructing a semantically rich representation space. In principle, ACR is algorithm-agnostic, allowing integration with any continuous control DRL algorithm. Our experiments on nine manipulation and navigation control tasks demonstrate the significant performance advantage of ACR compared to other action representation learning methods.

## 2 ACR: Action Chunk Representation

ACR considers solving a multi-step MDP, $\mathcal{M} = (\mathcal{S}, \mathcal{A}, k, \mathcal{C}, \bar{\mathcal{C}}, \mathcal{P}, \mathcal{R}, \gamma)$, where $\mathcal{S}$ is the state space, $\mathcal{A}$ is the action space, $k$ is the length of action , $\mathcal{C} = \prod_k \mathcal{A}$ is the action chunk space, $\bar{\mathcal{C}}$ is the representation space, $\mathcal{P}$ and $\mathcal{R}$ are the lifted transition function and reward function to accept action chunks, respectively. An action chunk can be defined as a multi-step action $c = \{a_t, a_{t+1}, ..., a_{t+k-1}\} \in \mathcal{C}$. The main idea of ACR is to learn an encoder $E_\phi(\bar{c}|c)$ that constructs a compact representation space for action chunks. The latent policy $\pi(\bar{c}|s)$ explores and learns in this space. It selects latent action chunks, which can be decoded to the original action space by the ACR decoder $D_\psi(c|\bar{c})$ to interact with the environment. The overview is shown in Fig.1, and the training process of ACR can be divided into two parts:

**Learning the representation space with Action Chunk Bisimulation Metric.** Our insight is that *action chunks with similar behaviors should cluster closely in the representation space, indicating those leading to the same transitions and rewards are equivalent in the associated multi-step MDP.* We extend the bisimulation metric [14] to measure the principled distance between action chunks:

**Definition 1** (Action Chunk Bisimulation Metric). Given a multi-step MDP $\mathcal{M}$ and a state $s_t$, a state-conditional action chunk bisimulation metric is a function $d_c$[1]: $\mathcal{S} \times \mathcal{C} \times \mathcal{C} \mapsto \mathbb{R}_{\geq 0}$ such that:

$$d(c_i, c_j | s_t) = \left| \mathcal{R}_{s_t}^{c_i} - \mathcal{R}_{s_t}^{c_j} \right| + \gamma \cdot W_2 \left( \mathcal{P}_{s_t}^{c_i}, \mathcal{P}_{s_t}^{c_j}; d_c \right) \tag{1}$$

---

[1]$d_c$ is a pseudometric for which $d(x, y) = 0 \Rightarrow x = y$.

where $W_2$ is the $2^{nd}$ Wasserstein distance between two distributions. Here, $\mathcal{R}_{s_t}^c$ represents the sum of cumulative discounted reward, i.e. $\sum_{i=0}^{k-1} \gamma^i \mathcal{R}_{s_{t+i}}^{a_{t+i}}$. $\mathcal{P}_{s_t}^c$ represents the distribution of $s_{t+k}$ after executing $c$ starting from $s_t$. The training process of ACR is shown in Fig. 2. To ensure action chunks in representation space satisfy the property $d(c_i, c_j | s_t) := |\bar{c}_i - \bar{c}_j|_1$, we train a Causal Transformer encoder $E_\phi(\bar{c}|c)$ to learn the representation space. We minimize self-supervised loss:

$$J(\phi) = \mathbb{E}_{s_t, c_i, c_j, \mathcal{R}_s^{c_i} \sim \mathcal{D},} \left( \|\bar{c}_i - \bar{c}_j\|_1 - \hat{d}(c_i, c_j \mid s_t) \right)^2, \tag{2}$$

$$\text{where } \hat{d}(c_i, c_j | s_t) = |\mathcal{R}_{s_t}^{c_i} - \hat{\mathcal{R}}(s_t, \bar{c}_j)| + \gamma \cdot W_2(\hat{\mathcal{P}}(\cdot | s_t, \bar{c}_i), \hat{\mathcal{P}}(\cdot | s_t, \bar{c}_j)). \tag{3}$$

$\hat{d}(c_i, c_j | s_t)$ represents an estimate of $d(c_i, c_j | s_t)$, $\hat{\mathcal{R}}$ and $\hat{\mathcal{P}}$ is the reward model and transition model, which are trained with the encoder separately. Please note that all inputs to the transition and reward model are stop-gradient. During the training of the decoder $D_\psi$, we fix the trained Transformer encoder and minimize the decoder objective function:

$$J(\psi) = \mathbb{E}_{c \sim \mathcal{D}} \left[ \|E_\phi(D_\psi(E_\phi(c))) - E_\phi(c)\|_2^2 + \lambda \|D_\psi(E_\phi(c))\|_2^2 \right] \tag{4}$$

The first term ensure that $D_\psi$ serves as a one-sided inverse of $E_\phi$, which means $E_\phi(D_\psi(\bar{c}) = \bar{c}$ but $D_\psi(E_\phi(c)) \neq c$. The second term guarantees that $D_\psi$ is the minimum-norm one-sided inverse of $E_\phi$, ensuring the validity of the decoded action chunk in the original action space.

**Optimizing the policy over the learned representation space.** ACR is algorithm-agnostic, allowing for integration with any continuous control DRL algorithm through minor modifications. Here, we use TD3 [15] as an example. The latent policy $\pi_\omega$ is trained in the representation space and should fully utilize the temporal information within the action chunk. Therefore, the TD3 double critic networks $Q_{\theta_{m=1,2}}$ additionally take as input the current step $i \in [0, k)$ within the executed action chunk. With a buffer of collected transition sample $b = (s_{t:t+k}, \bar{c}, r_{t:t+k-1}, i_{0:k-1})$, the critics are trained by Clipped Double Q-Learning:

$$L_{\text{CDQ}}(\theta_m) = \mathbb{E}_{b \sim \mathcal{D}}[(y - Q_{\theta_m}(s_{t+i}, \bar{c}, i)], \text{where}$$

$$y = \sum_{j=0}^{k-i-1} (\gamma^j r_{t+j}) + \gamma^{k-i} \min_{n=1,2} Q_{\bar{\theta}_n}(s_{t+k-i}, \pi_{\bar{\omega}}(s_{t+k-i}), 0) \tag{5}$$

$\bar{\theta}_{n=1,2}, \bar{\omega}$ are the the target network parameters. The actor considers the action chunk generated based on the initial state $s_t$ of each sample. We then update the actor as follows:

$$\nabla_\omega J(\omega) = \mathbb{E}_{s_t}[\nabla_\omega \pi_\omega(s_t) \nabla_{\pi_\omega(s_t)} Q_{\theta_1}(s_t, \bar{c}, 0)|_{\bar{c}=\pi_\omega(s_t)}] \tag{6}$$

Please note that as the latent policy improves the outdated representation may no longer reflect the same behavioral effects [16]. We propose two mechanisms to ensure the validity of the representation space: **Periodic Update** and **Adaptive Constraint**. Details are provided in Appendix A.

## 3 Experimental results

We evaluate ACR across nine continuous control tasks spanning three domains: 2DoF Arm Control [17], 7DoF Arm Control [18], and Maze Navigation [19] (See Appendix B.1 for more details), and aim to answer the following research questions (**RQs**): 1) Can combining ACR with DRL algorithms improve learning efficiency and performance? 2) Compared to other self-supervised action representation learning methods, what advantages does ACR offer? 3) What components and design choices of ACR contribute to the improved performance in our experiments?

**RQ1:** We combine ACR with three widely used DRL algorithms: DDPG, TD3, and SAC [20, 15, 21]. For a fair comparison, we use the default hyperparameters and architectures of them. As shown in Fig. 3 (Left), we observe significant performance improvements and faster convergence when ACR is combined with all three algorithms, especially in the challenging 7DoF robotic arm control tasks and sparse reward Maze navigation tasks. Notably, in Striker and Thrower, the original algorithms struggle to explore effectively due to the complexity of the action space, leading to unstable performance or even collapse. However, with ACR, the algorithms are able to learn stably. This evidence suggests that the action chunk representation space constructed by ACR captures the underlying structure of

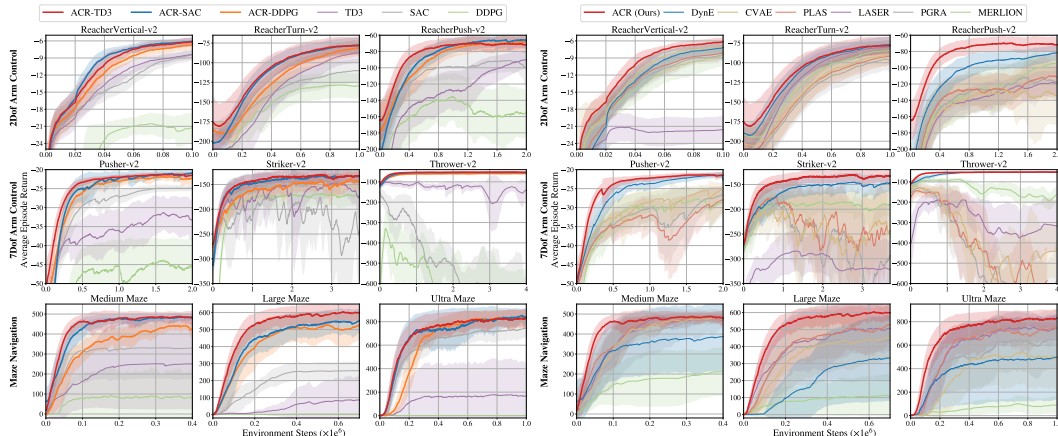

Figure 3: **Experimental results. Left:** Comparing ACR applied to three DRL algorithms against directly training them on the original action spaces. **Right:** Comparing the performance of ACR against different self-supervised action representation learning methods, each combined with TD3. The curve and shade denote the mean and a standard deviation over 5 random seeds.

the original action space, allowing the latent policy to explore and make decisions more effectively, thereby simplifying the tasks and significantly improving learning efficiency and performance.

**RQ2:** We combine ACR and other self-supervised action representation methods with TD3, training latent policies in their respective action representation spaces. These methods include: 1) Reconstruction: **PLAS** [12], 2) Forward Prediction: **CVAE**, **LASER** [9] and **DynE** [17], 3) Inverse Prediction: **PGRA** [8], 4) Bisimulation: **MERLION** [22] (see Appendix B.2.2 for details). As shown in Fig. 3 (Right), we observe that as task difficulty increases, other action representation learning methods suffer from performance collapse and unstable learning, while ACR maintains stable performance, especially in 7DoF arm control tasks. Additionally, we find that constructing action chunk representation space outperformed single-step action representation space, with ACR achieving significantly better performance and faster learning in the first two domains. These results demonstrate the intrinsic value of constructing an action chunk representation space.

**RQ3:** As shown in Fig. 4, we conduct ablation studies on 3 tasks: 1) **w/o ACB metric**: Removing the self-supervised learning objective for measuring action chunks distance (Equation 2) degrades ACR to a structure similar to conditional VAE, resulting in poor performance, indicating that the ACB metric is crucial for the representation space. 2) **w/o Transformer**: Replacing the Transformer encoder with an MLP leads to performance decline, highlighting the

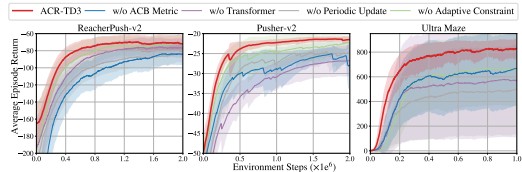

Figure 4: **Ablation studies.** The performance of ACR-TD3 relative to the version without the corresponding component or mechanism.

importance of temporal information within action chunks for high-quality representation. 3) **w/o Periodic Update**: The representation space is not continuously updated during policy training. 4) **w/o Adaptive Constraint**: The representation space range is fixed, similar to [12]. The experimental results show the absence of these two mechanisms harms learning performance and stability.

## 4 Conclusion

In this paper, we introduce ACR, a novel framework that uses the action chunk bisimulation metric as the self-supervised learning objective to construct a compact, low-dimensional, and decodable action chunk representation space for multi-step action, effectively capturing the semantic information of multi-step behavior. ACR is algorithm-agnostic, enabling integration with any continuous control DRL algorithm to enhance performance and learning efficiency. Our experiments demonstrate that ACR significantly outperforms other action representation learning methods, highlighting that the Action Chunk Bisimulation Metric captures richer semantic information. Additionally, the representation space constructed by ACR incorporates temporal information within action chunks, which is beneficial for uncovering the underlying structure of the action chunking space.

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

---

**Algorithm 1** ACR-TD3

---

1: Initialize actor $\pi_\omega$ and critic networks $Q_{\theta_1}, Q_{\theta_2}$ with random parameters $\omega, \theta_1, \theta_2$, and $\beta = 1$
2: Initialize components of ACR, including $E_\phi, D_\psi, \hat{\mathcal{P}}$ and $\hat{\mathcal{R}}$
3: Prepare replay buffer $\mathcal{D}_r = \{(s_{t:t+k}, c, r_{t:t+k-1})\}$ with random samples, where $c = \{a_t, \ldots, a_{t+k-1}\}$
4: # Stage 1: Learning the representation space with Action Chunk Bisimulation Metric
5: // Pre-training Transformer encoder
6: **for** $t = 1, \ldots, T_E$ **do**
7:     Sample a batch of $N$ tuples $(s_{t:t+k}, c_i, r_{t:t+k-1})$ and another batch of $c_j$ from $\mathcal{D}_r$
8:     Compute latent action chunk: $\bar{c}_i = E_\phi(c_i)$ and $\bar{c}_j = E_\phi(c_j)$
9:     Predict the transition distributions and the reward of $(s_t, c_j)$: $\hat{\mathcal{P}}(\cdot|s_t, \bar{c}_i), \hat{\mathcal{P}}(\cdot|s_t, \bar{c}_j)$ and $\hat{\mathcal{R}}(s_t, \bar{c}_j)$
10:     Compute the distance of $c_i$ and $c_j$: $\hat{d}(c_i, c_j|s_t)$         ▷ see Eq. 3
11:     Train Transformer encoder: $J(\phi)$         ▷ see Eq. 2
12:     Train transition and reward model: $J(\hat{\mathcal{P}}, \hat{\mathcal{R}}) = (\hat{\mathcal{P}}(\cdot|s_t, \bar{c}_i) - s_{t+k})^2 + (\hat{\mathcal{R}}(s_t, \bar{c}_i) - \mathcal{R}_{s_t}^{c_i})^2$
13: **end for**
14: // Pre-training the action chunk decoder
15: **for** $t = 1, \ldots, T_D$ **do**
16:     Sample a large batch of action chunks $c$ and compute latent space range $[-b, b]$
17:     Train action chunk decoder $J(\psi)$         ▷ see Eq. 4
18: **end for**
19: # Stage 2: Optimizing the policy over the learned representation space
20: **for** $t = 1$ to max environment step number $T_{\max}$ **do**
21:     // select latent action chunk in latent space and decode into original action space
22:     **if** $t \% k = 0$ **then**
23:         Observe state $s_t$ and select latent action chunk $\bar{c} = \pi_\omega(s_t) + \epsilon_e$, with $\epsilon_e \sim \mathcal{N}(0, \sigma b)$
24:         Decode latent action chunks using the decoder $c = D_\psi(\bar{c})$
25:         Execute action chunk $c$, observe $s_{t+1:t+k}$ and $r_{t:t+k-1}$
26:         Store $k$ experiences $\{(s_{t+i}, \bar{c}, a_{t+i}, r_{t+i}, s_{t+i+1})_i\}$ in $\mathcal{D}$, where $i \in [0, k-1]$
27:     **end if**
28:     Sample a batch of $N$ experiences from $\mathcal{D}$
29:     Update actor $\pi_\omega$ and critic $Q_{\theta_1}, Q_{\theta_2}$         ▷ see Eq. 5 and Eq. 6
30:     Update $\beta \leftarrow \beta - 4/T_{max}$ until $\beta = 0$
31:     // Update ACR at the intervals ($I$)
32:     **if** $t \% I = 0$ **then**
33:         Sample $\beta N$ experiences from $\mathcal{D}_r$ and $(1 - \beta)N$ experiences from $\mathcal{D}$
34:         Update $\phi, \psi, \hat{\mathcal{P}}$ and $\hat{\mathcal{R}}$
35:     **end if**
36: **end for**

---

# A   Additional Details of ACR

## A.1   Details of Periodic Update and Adaptive Constraint

**Periodic Update**   We default to using samples generated by a random policy to learn the representation space before training the latent policy, as random samples can cover the environmental dynamics during the initial stage of exploration, thereby accelerating the early learning rate of the latent policy. However, as the latent policy improves, outdated representations may no longer reflect the same behavioral effects [7]. To address this issue, we periodically update the ACR during training to adjust the distribution of representation space continuously. Specifically, we introduce a parameter $\beta$, initially set to 1, which decays to 0 when reaching $1/4$ of the maximum environment steps. At every interval $I$, we update the ACR using a proportion of $\beta$ random samples and $1 - \beta$ real samples.

**Adaptive Constraint**   The range of the representation space constrains the output range of the latent policy, ensuring that latent action chunks remain meaningful within the current latent space. Similar evidence has been found in [12]. To address this issue, at the end of each ACR adjustment, we sample a large batch of action chunks $c$ and obtain latent action chunks using the updated encoder $\bar{c} = E_\theta(c)$. We then calculate the mean $\mu_{\bar{c}}$ and standard deviation $\sigma_{\bar{c}}$, and compute the max representation space value $b = max|\frac{\bar{c} - \mu_{\bar{c}}}{\sigma_{\bar{c}}}|$. This constrains the current representation space range to a reasonable interval $[-b, b]$, i.e., the latent policy uses tanh activation and scales the output by the value $b$. Compared to previous methods with explicit constraints, our approach is more flexible and easier to optimize.

# B Details of Experimental Setup

## B.1 Environments

As shown in Fig.5, the first domain in this paper is **2DoF Arm Control** provided by [17] which is based on the `Reacher-v2` MuJoCo [18] task from OpenAI Gym [23]. This domain includes three distinct tasks: `ReacherVertical`, a standard target-reaching task; `ReacherTurn`, inspired by the DeepMind Control Suite's `Finger Turn` environment, where a 2-link Reacher robot must rotate a spinner to a specified random location; and `ReacherPush`, derived from the `Stacker` environment, requiring the Reacher to push a block to a randomly generated target position. The rationale behind selecting this domain lies in the constrained nature of the 2DoF arm's action space. Specifically, in tasks like `ReacherPush`, the arm's actions are only meaningful when interacting with the brown box in the lower half region. Actions in the upper region are effectively inconsequential, creating a structured action space that is particularly suitable for evaluating ACR.

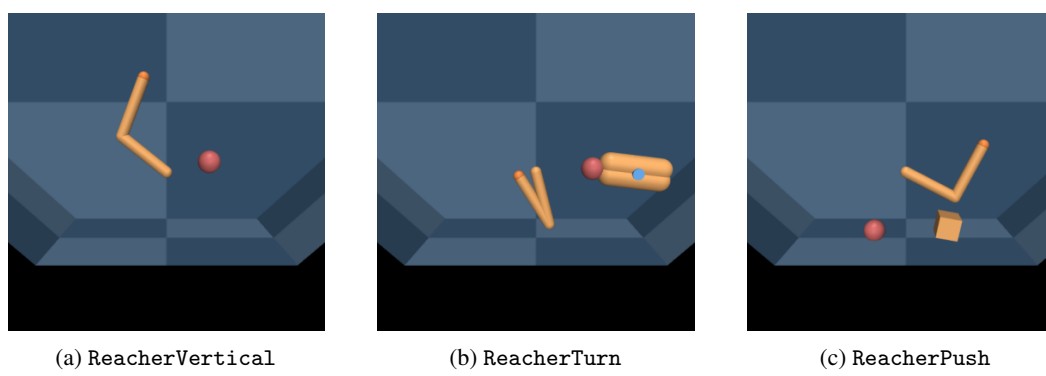

(a) `ReacherVertical`     (b) `ReacherTurn`     (c) `ReacherPush`

Figure 5: **2DoF Arm Control**. `ReacherVertical` requires the agent to move the tip of the arm to the red dot. `ReacherTurn` requires the agent to turn a rotating spinner (dark red) so that the tip of the spinner (gray) is close to the target point (red). `ReacherPush` requires the agent to push the brown box onto the red target point. The initial state of the simulator and the target point are randomized for each episode. In each task the rewards are dense and there is a penalty on the norm of the actions. Every episode consists of 100 steps.

As shown in Fig.6, the second domain is **7DoF Arm Control** from OpenAI Gym [23], including `Pusher-v2`, `Striker-v2`, and `Thrower-v2`. These tasks are considerably challenging because the target points are randomly generated, and it is impractical for the robotic arm to explore the original action space's invalid regions (e.g., rapidly moving the arm up or down). We propose ACR, which constructs action chunking representation space, making the exploration for policy easier and more efficient.

As shown in Fig.7, the third domain is Maze Navigation, including `Medium Maze` and `Large Maze` are versions of the Maze2D tasks from D4RL [19]. Additionally, we have constructed a more complex `Ultra Maze`. In these three tasks, due to the sparse reward signal, exploring within the effective action space is crucial.

## B.2 Baselines

### B.2.1 Continuous-control DRL Algorithms

For TD3 and DDPG, we use the official implementations (`https://github.com/sfujim/TD3`). For SAC, we implement it with reference to the hyperparameters from the official implementation (`https://github.com/haarnoja/sac`). For details of the combination of ACR and the three DRL algorithms, please refer to Appendix C.

### B.2.2 Action Representation Methods

In this section, we introduce other self-supervised action representation learning methods that have been proven effective in both online and offline settings, though our focus is on the online setting. We

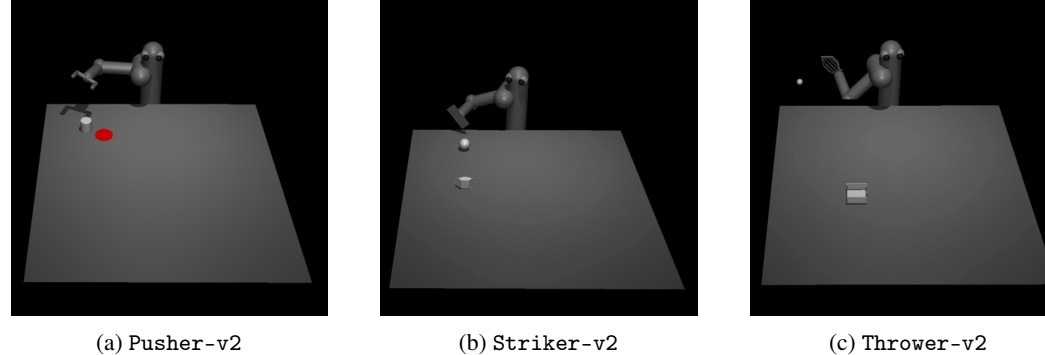

(a) `Pusher-v2`     (b) `Striker-v2`     (c) `Thrower-v2`

Figure 6: **7DoF Arm Control**. `Pusher-v2` requires the agent to use a C-shaped end effector to push a puck across the table onto a red circle. `Striker-v2` requires the agent to use a flat-end effector to hit a ball so that it rolls across the table and reaches the goal. `Thrower-v2` requires the agent to throw a ball to a target using a small scoop. Each task features dense rewards, and every episode consists of 100 steps.

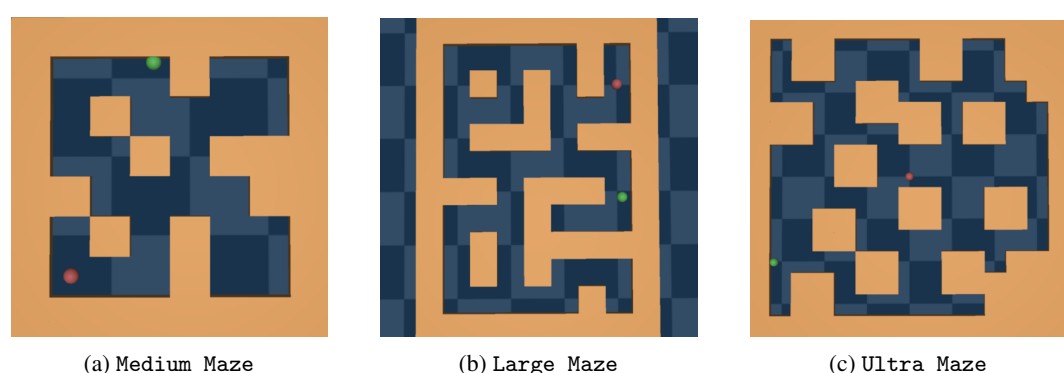

(a) `Medium Maze`     (b) `Large Maze`     (c) `Ultra Maze`

Figure 7: **Maze Navigation**. `Medium Maze`, `Large Maze` and `Ultra Maze` require the agent (green ball) to navigate through mazes of varying complexity until it reaches the target point (red ball). For each episode, the position of the target point is fixed while the position of the agent is randomly generated. The reward signal for each task is sparse, and each episode consists of 600, 800, and 1000 steps, respectively.

provide implementation details for each baseline used in our experiments, along with any reasonable and necessary modifications made to ensure fair comparison.

**Reconstruction** PLAS [12] uses a conditional VAE [24] on offline datasets to construct the single-step state-conditioned action representation space, then employs the VAE decoder to reconstruct the action. We use the official implementation of PLAS (`https://github.com/Wenxuan-Zhou/PLAS`) but focus on the online setting. Specifically, we pre-train the VAE model with samples generated by a random policy and allow the latent policy to explore and update online over this space. The hyperparameters are presented in Table 1.

**Forward Prediction** Forward prediction-based self-supervised learning objectives are widely used in action representation methods. We introduce the following three methods based on their learning objectives and network structures: 1) **CVAE** can be viewed as a simplified version of **HyAR** [7] without the embedding table for discrete action. HyAR focuses on hybrid action space settings, using an embedding table and a conditional VAE to capture dependency between discrete and continuous action. The decoder then reconstructs the continuous action, and an additional cascaded network predicts the state residual caused by the hybrid action. The embedding table is removed in **CVAE**, allowing the model to predict the state residual caused by continuous action. The hyperparameters and network structure are presented in Table 2. 2) **LASER** [9] is similar to CVAE but replaces the cascaded network with an additional transition model to predict the next state. The learning objective shifts to action reconstruction and state transition prediction.

Table 1: Hyperparameters of PLAS.

| Hyperparameter | Value |
|---|---|
| Encoder hidden layers | 3 |
| Encoder input dimension | state dim + action dim |
| Latent dimension | action dim · 2 |
| Decoder hidden layers | 3 |
| Decoder input dimension | state dim + (action dim · 2) |
| Layer width | 256 |
| Activation | ReLU |
| Batch size | 100 |
| Optimizer | Adam |
| Learning rate | $10^{-4}$ |

We modified the PLAS code to implement LASER. The hyperparameters and network structure are presented in Table 3. 3) **DynE** [17] uses a conditional VAE to construct a multi-step action representation space, with the VAE decoder predicting the state distribution after executing the multi-step actions. To enable the latent policy to interact with the environment over the representation space, DynE additionally learns an action decoder that decodes the high-level action into multi-step actions. We used the official implementation (`https://github.com/dyne-submission/dynamics-aware-embeddings`).The hyperparameters and network structure are presented in Table 4.

Table 2: Hyperparameters and network structure of CVAE.

| Model Component | Layer (Name) | Structure (Value) |
|---|---|---|
| VAE encoder | Fully Connected (input) | (state dim + action dim, 256) |
| | Activation | ReLU |
| | Fully Connected | (256, 256) |
| | Activation | ReLU |
| | Fully Connected (mean) | (256, latent dim) |
| | Fully Connected (logstd) | (256, latent dim) |
| VAE decoder | Fully Connected (latent) | (state dim + latent dim, 256) |
| | Activation | ReLU |
| | Fully Connected | (256, 256) |
| | Activation | ReLU |
| | Fully Connected (reconstruction) | (256, action dim) |
| | Fully Connected | (256, 256) |
| | Activation | ReLU |
| | Fully Connected (prediction) | (256, state dim) |
| | Latent dimension | action dim |
| | Batch size | 100 |
| | Optimizer | Adam |
| | Learning rate | $10^{-4}$ |

**Inverse Prediction**  **PGRA** [8] embeds single-step discrete action influence on environmental dynamics to construct the action representation space, enabling generalization over large discrete action sets and allowing latent policy learning over the representation space. We implement PGRA's action representation network with an encoder as a 3-layer MLP, which encodes $s_t$ and $s_{t+k}$ to the latent space $z$, and a decoder, structured as a 2-layer MLP, which decodes the latent action into the original action space action $\hat{a}$. The action representation network is optimized by minimizing the mean squared error between the original action $a$ and the predicted action $\hat{a}$. The hyperparameters are presented in Table 5.

**Bisimulation**  **MERLION** [22] focuses on the offline setting for large discrete action spaces, extending the bisimulation metric to single-step actions to measure behavioral relation between actions in the dataset. We implement this baseline for continuous action spaces in online setting. The hyperparameters and network structure are presented in Table 6.

Table 3: Hyperparameters and network structure of LASER.

| Model Component | Layer (Name) | Structure (Value) |
|---|---|---|
| VAE encoder | Fully Connected (input) | (state dim + action dim, 256) |
| | Activation | ReLU |
| | Fully Connected | (256) |
| | Activation | ReLU |
| | Fully Connected (mean) | (256, latent dim) |
| | Fully Connected (logstd) | (256, latent dim) |
| VAE decoder | Fully Connected (latent) | (state dim + latent dim, 256) |
| | Activation | ReLU |
| | Fully Connected (reconstruction) | (256, action dim) |
| Transition model | Fully Connected (latent) | (state dim + latent dim, 256) |
| | Normalization | Layer Normalization |
| | Activation | ReLU |
| | Fully Connected (prediction) | (256, state dim) |
| | Latent dimension | 4 |
| | Batch size | 100 |
| | Optimizer | Adam |
| | Learning rate | $10^{-4}$ |

Table 4: Hyperparameters and network structure of DynE.

| Model Component | Layer (Name) | Structure (Value) |
|---|---|---|
| VAE encoder | Fully Connected (input) | (action dim $\cdot$ k, 256) |
| | Activation | SeLU |
| | Fully Connected | (256) |
| | Activation | SeLU |
| | Fully Connected (mean) | (256, latent dim) |
| | Fully Connected (logstd) | (256, latent dim) |
| VAE decoder | Fully Connected (latent) | (state dim + latent dim, 256) |
| | Activation | SeLU |
| | Fully Connected (latent) | (256, 256) |
| | Activation | SeLU |
| | Fully Connected (prediction) | (256, state dim) |
| Action decoder | Fully Connected (latent) | (latent dim, 256) |
| | Activation | SeLU |
| | Fully Connected | (256, 256) |
| | Activation | SeLU |
| | Fully Connected (decode) | (256, action dim $\cdot$ k) |
| | Activation | Tanh |
| | Multi-step action length $k$ | 4 |
| | Latent dimension | action dim |
| | Batch size | 100 |
| | VAE Optimizer | Adam |
| | action decoder Optimizer | Adam |
| | Learning rate | $10^{-4}$ |

# C  Implementation Details

## C.1  Actor and Critic Networks

When combining ACR with DDPG, TD3, and SAC, we modified the network architecture of the algorithms to enable the policy to explore and train over the action chunking representation space. Specifically, we changed the output dimension of the actor network from the original single-step action dimension to the *latent dimension*, representing the policy's output of latent action chunks. For the Critic network, we added an 1-dimension $i$ to the state and latent dimension, representing the current step $i \in [0, k)$ within the executed action chunk. In contrast, when other single-step action representation methods are combined with DRL algorithms, the input dimension $i$ is not required.

Table 5: Hyperparameters of PGRA.

| Hyperparameter | Value |
|---|---|
| Encoder hidden layers | 3 |
| Encoder input dimension | state dim $\cdot$ 2 |
| Latent dimension | action dim |
| Decoder hidden layers | 2 |
| Layer width | 256 |
| Activation | ReLU |
| Batch size | 100 |
| Optimizer | Adam |
| Learning rate | $10^{-4}$ |

Table 6: Hyperparameters and network structure of MERLION.

| Model Component | Layer (Name) | Structure (Value) |
|---|---|---|
| Action encoder | Fully Connected (input) | (state dim + action dim, 256) |
| | Activation | ReLU |
| | Fully Connected | (256) |
| | Activation | ReLU |
| | Fully Connected (latent) | (256, latent dim) |
| Transition model | Fully Connected (latent) | (state dim + latent dim, 256) |
| | Normalization | Layer Normalization |
| | Activation | ReLU |
| | Fully Connected (prediction) | (256, state dim) |
| Reward model | Fully Connected (input) | (state dim + latent dim, 256) |
| | Activation | ReLU |
| | Fully Connected | (256) |
| | Activation | ReLU |
| | Fully Connected (latent) | (256, 1) |
| | Activation | Tanh |
| | Multi-step action length $k$ | 4 |
| | Latent dimension | action dim |
| | Batch size | 100 |
| | VAE Optimizer | Adam |
| | action decoder Optimizer | Adam |
| | Learning rate | $10^{-4}$ |

## C.2 Network structures of ACR

Our implementation of the Causal Transformer encoder $E_\phi$ is based on the implementation of the Decision Transformer [25] available at `https://github.com/kzl/decision-transformer`. In our implementation, we encode only the action chunk $c$ without conditioning on the state $s_t$. Additionally, the relative temporal information $i \in [0, k)$ within each action chunk is embedded and added to the action tokens. As a result, the encoder outputs $k$ feature vectors. We take only the last vector as the latent action chunk because it encapsulates both the temporal information across $k$ steps and the semantic information. The reward model $\hat{\mathcal{R}}$ is parameterized as a 3-layer MLP with ReLU activations after each layer except the last. For the transition model $\hat{\mathcal{P}}$, please refer to the implementation of MERLION. Both $\hat{\mathcal{R}}$ and $\hat{\mathcal{P}}$ take the current state $s_t$ and latent action chunk $\bar{c}$ as inputs to predict the cumulative discounted reward and the state transition $s_{t+k}$, respectively.

Table 7: Hyperparameters of ACR.

| Training Stage | Hyperparameter | Value |
|---|---|---|
| Pre-training $E_\phi$ | Number of layers | 2 |
| | Number of attention heads | 1 |
| | Hidden dim | 128 |
| | Activation | ReLU |
| | Dropout | 0.1 |
| | Weight decay | $10^{-4}$ |
| | $E_\phi$ optimizer | AdamW |
| | Betas of AdamW | (0.9, 0.99) |
| | $\hat{\mathcal{P}}$ and $\hat{\mathcal{R}}$ optimizer | Adam |
| | Learning rate | $10^{-4}$ |
| | Batch size $N$ | 100 |
| | Action chunk length $k$ | 8 Thrower 4 Other |
| | Pre-training steps $T_E$ | $10^4$ |
| | Random buffer size $\mathcal{D}_r$ | $10^4$ |
| Pre-training $D_\psi$ | Hidden dim | 256 |
| | Layer | 3 |
| | Activation | ReLU |
| | Coefficient $\lambda$ | $10^{-4}$ |
| | Pre-training steps $T_D$ | $10^4$ |
| Policy training | Discount Factor $\gamma$ | 0.99 |
| | Buffer size $\mathcal{D}$ | $10^5$ |
| | Update intervals $I$ | 100 steps |

