# OpenReview forum: "Self-Supervised Bisimulation Action Chunk Representation for Efficient RL"
_NeurIPS.cc/2024/Workshop/SafeGenAi — SafeGenAi Poster_

### Official Review · Reviewer_xSoZ · 2024-10-08

**Rating:** 6
**Confidence:** 4

**Review:**

**Strength**
1. This paper explores the intriguing topic of action chunking in reinforcement learning and proposes an effective method, ACR, which uncovers the underlying structure of the action chunk space to enhance the efficiency of RL.
2. This paper presents comprehensive experimental results that demonstrate the efficiency of the proposed method.

**Weakness**
1. This paper centers on the exploration of the underlying structure of the action chunk space to enhance the efficiency of RL. While this is an important and valuable area of research, it seems somewhat misaligned with the primary focus of the workshop, which is centered on 'Safe Generative AI'. The absence of a direct connection to safety in generative models or AI systems raises questions about the paper's relevance to the workshop's intended theme.
2. ACR encodes action chunks using a Transformer, which captures the temporal structure and learns a latent representation space where action chunks with similar bisimulation behavior semantics are positioned near one another. While this offers a promising approach, a more detailed explanation of how ACR specifically enhances the effectiveness of action chunking in reinforcement learning is necessary. In particular, elaborating on how this latent space contributes to improved decision-making and learning efficiency would provide valuable insights. Additionally, providing theoretical support to substantiate these claims would significantly strengthen the validity of the approach.

**Question**
1. Figure 2 requires adjustment, as the reward model's $R_{s_t}^{c_i}$ is derived from $D_r$. Therefore, $R_{s_t}^{c_i}$ should be depicted within $D_r$ to accurately reflect its source.
2. The pseudometric $d_c$, as introduced in Equation (1), requires further explanation. Providing additional clarification on its definition, properties, and how it is applied within the context of the paper would be provided.
3. Experiments comparing ACR-DRL algorithms with other strong baselines are expected, particularly where ACR is combined with other robust DRL algorithms, such as CQL and IQL, to demonstrate its effectiveness.
4. Theoretical support for ACR is expected to strengthen the credibility of the method, making it more convincing.

---

### Official Review · Reviewer_mWfa · 2024-10-08
**Review of this paper**

**Rating:** 7
**Confidence:** 4

**Review:**

## Summary
This paper presents a framework that uses the action chunk bisimulation metric as the self-supervised learning objective to construct an action chunk representation space for multi-step decision-making. Experiments on various tasks demonstrate the superior performance of the proposed method.
## Strengths
- Action chunk representation is novel, and I believe it is promising for future imitation learning research.
- The method seems to be sound, and the writing is good.
- Extensive baselines are compared in experiments.

---

### Official Review · Reviewer_PZrh · 2024-10-09
**proposed bisimulation action chunk representation to improve deep RL algorithm performance**

**Rating:** 6
**Confidence:** 3

**Review:**

The authors propose to use action chunk bisimulation metric to learn a representation space and then optimize the policy over the learned representation space. They propose periodic update and adaptive constraint, two mechanisms to ensure the validity of the learnt representation space. Empirical results demonstrate that the learnt representation combined with deep RL algorithms can improve performance.

Strength: The authors provided answers for a few important research questions in experimental results. They not only compared with DRL methods without ACR, but also compared with other self-supervised learning methods. They did ablations on different components of ACR and analyzed which parts lead to improved performance.

Weakness: it is not clear from main results (Figure 3 right) that the proposed method outperforms other self-supervised action representation learning methods. The error bars are overlapping for most tasks, and the mean return is about the same as the best existing method on about half of the tasks. One could make an observation that the proposed method is indeed more robust as each of the baselines would substantially underperform on at least one task. It would improve the paper to show paired t-test results to demonstrate statistical significance.

Additional comments: why did the authors choose to use bisimulation metric instead of other metrics that could potentially measure the distance between action chunks? It would be nice to provide some insights.